# Developmental Regulation of the Expression of Amaryllidaceae Alkaloid Biosynthetic Genes in *Narcissus papyraceus*

**DOI:** 10.3390/genes10080594

**Published:** 2019-08-07

**Authors:** Tarun Hotchandani, Justine de Villers, Isabel Desgagné-Penix

**Affiliations:** 1Department of Chemistry, Biochemistry and Physics, Université du Québec à Trois-Rivières, 3351 boulevard des Forges, Trois-Rivières, QC G9A 5H7, Canada; 2Plant Biology Research Group, Trois-Rivières, QC G9A 5H7, Canada

**Keywords:** alkaloids, Amaryllidaceae, transcriptome, specialized metabolism, developmental stage, gene expression, *N. papyraceus*, HPLC, RT-qPCR, biosynthesis

## Abstract

Amaryllidaceae alkaloids (AAs) have multiple biological effects, which are of interest to the pharmaceutical industry. To unleash the potential of Amaryllidaceae plants as pharmaceutical crops and as sources of AAs, a thorough understanding of the AA biosynthetic pathway is needed. However, only few enzymes in the pathway are known. Here, we report the transcriptome of AA-producing paperwhites (*Narcissus papyraceus* Ker Gawl). We present a list of 21 genes putatively encoding enzymes involved in AA biosynthesis. Next, a cDNA library was created from 24 different samples of different parts at various developmental stages of *N. papyraceus*. The expression of AA biosynthetic genes was analyzed in each sample using RT-qPCR. In addition, the alkaloid content of each sample was analyzed by HPLC. Leaves and flowers were found to have the highest abundance of heterocyclic compounds, whereas the bulb, the lowest. Lycorine was also the predominant AA. The gene expression results were compared with the heterocyclic compound profiles for each sample. In some samples, a positive correlation was observed between the gene expression levels and the amount of compounds accumulated. However, due to a probable transport of enzymes and alkaloids in the plant, a negative correlation was also observed, particularly at stage 2.

## 1. Introduction

Plants produce specialized metabolites having a wide range of biological activities. One class of specialized metabolites, the alkaloids, are mainly characterized by the occurrence of at least one nitrogen atom in a heterocyclic ring. Since alkaloids are often toxic, they are believed to play an important role in the defense against herbivores and pathogens [1]. The toxic properties (cytotoxic, cytostatic, anti-microbial, etc.) of alkaloids make them suitable for use, at low doses, as medical drugs. For example, the alkaloid vinblastine is used as an anticancer drug, whereas morphine and codeine are two well-known analgesic alkaloids [2].

One family of plants known to produce a wide variety of alkaloids with promising medicinal potential is the Amaryllidaceae family [3]. It includes ornamental species such as daffodils, jonquils and snowdrops. More than 600 alkaloids synthesized by the Amaryllidaceae family have been identified so far [4]. However, only one Amaryllidaceae alkaloid (AA), galanthamine, is currently available as a commercial drug, marketed under the names Reminyl and Razadyne [5]. Owing to its acetylcholinesterase inhibitory property, it is used to treat the symptoms of Alzheimer’s disease [6,7]. Many other AAs are being studied for eventual clinical use. For example, lycorine and narciclasine exhibit anticancer effects. Lycorine is a powerful inhibitor of cell growth and cell division [8,9], whereas narciclasine has been shown to be effective against human glioblastoma multiform tumors in preclinical animal models [10].

Unfortunately, large-scale production of AAs is difficult and expensive. Firstly, many AAs exist only in trace amounts in their native plants, and growing a large number of Amaryllidaceae plants in order to obtain reasonable amounts of AAs is environmentally unsustainable. Secondly, total chemical synthesis can help alleviate the demands for a limited number of AAs, but since these compounds have complex structures, their synthesis is quite challenging. Therefore, the biotechnological methods based on in vitro cultures [11,12,13,14,15,16,17] or the genetic engineering of plants or microorganisms offers promising potential alternative sources for the production of high concentrations of AAs and would be a more environmentally sustainable and cost-effective approach. This, however, requires a better understanding of the AA biosynthetic pathway.

Early feeding experiments using radiolabelled precursors led to the biochemical elucidation of the initial steps in AA biosynthesis [13,14,18,19,20,21,22,23]. These studies suggest that, despite their large structural diversity, all AAs are derived from a common metabolic precursor, norbelladine (Figure 1). Furthermore, experimental studies coupled with analogies with other plant alkaloid pathways showed that the central compound norbelladine is formed from the condensation of 3,4-dihydroxybenzaldehyde (3,4-DHBA) and tyramine (Figure 1) [24].

The biosynthesis of AAs begins with the two amino acids phenylalanine and tyrosine. A series of biochemical reactions eventually leads to the production of the more than 600 AAs known to date [25]. In the pathway, all the compounds synthesized from these two amino acids, leading to the formation of norbelladine, are referred to as the AA precursor compounds. On the other hand, the compounds produced from norbelladine, leading to the formation of AAs, are referred to as the AA intermediate compounds. At present, the AA precursors *trans*-cinnamic acid, *p*-coumaric acid, caffeic acid, 3,4-DHBA, tyramine, etc. are known, and most of the biosynthetic genes encoding enzymes putatively involved in their production have been identified in other plants (Figure 1).

A central dogma of medicinal chemistry and chemical biology is that compounds with similar structures have similar activities [26]. Hence, similar structures suggest similar metabolic pathways involving similar biosynthetic genes. Since compounds similar to the AA precursors and intermediates are present in plants belonging to other alkaloid-producing plant families, such as Papaveraceae and Solanaceae, where the biosynthetic genes are known, it has been proposed that genes homologous to the known biosynthetic genes may be involved in AA biosynthesis. Therefore, the biosynthetic genes *PAL* (coding for phenylalanine ammonia lyase), *C4H* (coding for cinnamate 4-hydroxylase (CYP73A1)) and *C3H* (coding for coumarate 3-hydroxylase (CYP98A3)) have been proposed, so far, as being responsible for the production of 3,4-DHBA from phenylalanine (Figure 1). Regarding caffeic acid, it is uncertain whether it is produced directly from *p*-coumaric acid involving only *C3H* or whether it is produced via the sequence of reactions, shown in the grey shaded area in the pathway, involving the additional proposed AA biosynthetic genes *4CL* (coding for 4-coumarate CoA ligase) and *HCT* (coding for hydroxycinnamoyl transferase). Concerning tyramine, it is assumed *TYDC* (coding for tyrosine decarboxylase) is responsible for its production from tyrosine [27]. Recently, *NBS* (coding for norbelladine synthase) was identified from a transcriptome database and is responsible for the production of norbelladine from the condensation of 3,4-DHBA and tyramine [24]. Regarding the production of AA intermediates, the biosynthetic genes *N4OMT* (coding for norbelladine 4′-*O*-methyltransferase), *CYP96T1* (coding for cytochrome P450 monooxygenase 96T1) and *NorRed* (coding for noroxomaritidine reductase) have been confirmed [28,29,30]. Therefore, there are six proposed (*PAL*, *C4H*, *C3H*, *4CL*, *HCT* and *TYDC*) and four confirmed (*NBS*, *N4OMT*, *CYP96T1* and *NorRed*) AA biosynthetic genes in the pathway at the moment. The reason behind the great diversity of AAs is primarily due to the three different types of oxidative phenol coupling reactions (para-ortho’, ortho-para’ and para-para’), which convert 4′-*O*-methylnorbelladine into myriad alkaloids (Figure 1).

In the present work, we attempt to gain a better understanding of the involvement of the AA biosynthetic genes in the production of AAs, and to study spatial and temporal differences regarding AA biosynthesis in the plant. To do this, we analyzed different parts of the Amaryllidaceae species *Narcissus papyraceus* Ker Gawl at various developmental stages. This species was selected since, among the Amaryllidaceae species, it can be easily grown indoors in a controlled environment. Differences in AA accumulation and biosynthetic genes’ expression have also been reported for *Narcissus* species [25,31,32,33,34]. A deeper understanding of the molecular mechanisms involved in AA biosynthesis will support breeding efforts to produce cultivars of the Amaryllidaceae species with enhanced AA production capabilities. In addition, it will pave the way for the successful metabolic engineering of microbial systems for the production of valuable AAs.

## 2. Materials and Methods

### 2.1. Chemicals

Anhydrous ethyl alcohol was purchased from Commercial Alcohols (Brampton, ON, Canada). Ammonium acetate (anhydrous, ACS reagent grade) was obtained from MP Biomedicals (Solon, OH, USA). Ammonia (ca. 7N solution in methanol) was ordered from ACROS Organics (www.acros.com). Agarose (protein electrophoresis grade, high gelling temperature); Tris base (for molecular biology); EDTA, disodium salt dihydrate; sulfuric acid; acetic acid, glacial (certified ACS); methanol (HPLC grade); chloroform (HPLC grade) and acetonitrile (HPLC grade) were purchased from Fisher Scientific (Fair Lawn, NJ, USA). The alkaloids narciclasine and galanthamine were purchased from Tocris Bioscience (Bristol, UK), whereas lycorine and papaverine were obtained from Sigma-Aldrich (Oakville, ON, Canada).

### 2.2. Plant Material

Twenty-four *N. papyraceus* bulbs were purchased from Veseys (York, PE, Canada). Three bulbs were kept aside and 21 were planted using AGRO MIX G6 potting soil (Fafard, Saint-Bonaventure, QC, Canada), which had been autoclaved for 40 min. The plants were kept at room temperature with exposure to tube lighting for 16 h daily until being harvested. The plants were watered when necessary to keep the soil moist. They were harvested at each developmental stage (Figure 2) over an 85-day period (Table A1).

The three unplanted bulbs represented the first stage of development. A total of 10 g of bulbs was extracted and stored at −80 °C. Among the 21 planted bulbs (for the five subsequent developmental stages), three to four plants were collected for each stage and the same amount (10 g) was extracted from different parts of the plant and stored at −80 °C. Up to five parts (bulb, roots, leaves, stem and flowers) being studied at six developmental stages resulted in a total of 24 plant samples.

### 2.3. RNA Extraction, Next-Generation Illumina Sequencing and *de novo* Transcriptome Assembly

Two grams of the stage 1 bulb sample were ground in liquid nitrogen with a mortar and pestle and transferred to pre-chilled 50 mL tubes to proceed with the CTAB (cetrimonium bromide) method for total RNA extraction as described by Singh and Desgagné-Penix (2017) [25]. The integrity of the RNA was verified using the NanoVue spectrophotometer (GE Healthcare Life Sciences, www.gelifesciences.com) and bioanalyzer. Bioanalysis of the RNA gave an RNA integrity number of 8.1 with a 28S/18S value of 1.54, thus confirming the suitable integrity and quality of the RNA. The pipeline used for trimming and de novo transcriptome assembly has been previously described [35] and is mostly based on the Trinity assembly software suite [36].

Briefly, the mRNA was converted into a cDNA library and sequenced using Illumina’s HiSeq 2000 sequencing system, PE100 paired-ends, at McGill University and Génome Québec Innovation Centre (Montreal, QC, Canada). Raw paired reads were trimmed from the 3′-end to obtain a Phred score of at least 30. Illumina’s sequencing adapters were removed, maintaining 50 bp of the minimum read length to obtain the surviving paired reads. Trimming and clipping was done using Trimmomatic (http://www.usadellab.org/cms/index.php?page=trimmomatic) [37] for quality filtering to obtain clean reads. Once the surviving pair data was generated, Trinity normalization was performed to eliminate redundant reads in datasets without affecting its k-mer content [38]. De novo assembly of the cleaned and normalized reads was done using the Trinity assembler (https://github.com/trinityrnaseq/trinityrnaseq/wiki) [36]. The final unigenes obtained were functionally annotated using Trinotate (http://trinotate.github.io/). To quantify the gene transcript abundance, the raw RNA-Seq reads were mapped to the assembled transcripts applying Bowtie [39] using default parameters. The gene transcript abundance was calculated as fragments per kilobase of transcript per million mapped reads (FPKM) using the RSEM package [40].

### 2.4. Primer Design

To design primer pairs specific to a region of a particular AA biosynthetic gene, three to four partial or complete homologous mRNA sequences from species closely related to *N. papyraceus* were selected from the Nucleotide database of the National Center for Biotechnology Information (NCBI, www.ncbi.nlm.nih.gov). The similarity of the selected sequences was verified by performing a multiple sequence alignment using CLUSTALW (Kyoto University Bioinformatics Center, www.genome.jp/tools-bin/clustalw). From the aligned sequences, the longest one having the least number of gaps was selected. This sequence was queried against our newly assembled *N. papyraceus* transcriptome using NCBI’s megablast BLASTN program. Among the transcripts selected by BLASTN, the one with the highest Query cover and Ident values was used in the designing of primers with the online PrimerQuest tool (Integrated DNA Technologies, www.idtdna.com/PrimerQuest). Primer pairs producing short amplicons of 100–300 bp (a requirement for reverse transcription quantitative real-time PCR (RT-qPCR) experiments) were chosen. Primers were also designed for the gene *HIS*, encoding histone, used as a reference gene in the RT-qPCR experiments. The primers are listed in Table A2.

### 2.5. cDNA Synthesis

A mass of 0.1 g of each plant sample was ground in liquid nitrogen with a mortar and pestle. RNA was extracted from each ground-up plant sample using GENEzol TriRNA Pure Kit (Geneaid, www.geneaid.com). The RNA sample obtained was diluted with water to a concentration of 100 ng/μL. Ten µL (or 1 µg) of RNA was then immediately used for reverse transcription or stored at −80 °C. Prior to reverse transcription, the purity of the RNA was verified using the NanoVue spectrophotometer. A ratio of the absorbance at 260 and 280 nm (A_260/280_) greater than 1.8, and the ratio A_260/230_ greater than 2.0 are considered to be suitable for gene expression measurements [41]. cDNA synthesis was carried out using Omniscript Reverse Transcription Kit (QIAGEN, www.qiagen.com) with the aid of oligo-dT. The cDNA created was stored at −20 °C.

### 2.6. Quantitative Real-Time Reverse Transcription PCR (RT-qPCR)

RT-qPCR was performed on triple technical replicates of each plant sample using the CFX Connect Real-Time System (Bio-Rad, www.bio-rad.com). The reaction mixtures were prepared using SensiFAST SYBR Lo-ROX Kit (Bioline, www.bioline.com/sensifast). The provided protocol was followed using 1 µL of template cDNA (100 ng/µL). For each gene, the primers used and the temperature for the annealing/extension step of the reaction are listed in Table A2. The reaction conditions were 95 °C for 3 min, followed by 40 cycles of denaturation at 95 °C for 10 s each and annealing/extension for 30 s. The last steps of the reaction were 95 °C for 10 s, 65 °C for 5 s and 95 °C for 5 s. The threshold cycle (C_T_) value of each gene was normalized against the C_T_ value of *HIS*, the reference gene. Mean C_T_ values, calculated from the technical triplicates, were used for relative quantitative gene expression analysis involving the comparative C_T_ method [42]. The statistical error was calculated using the method stated in the CFX Connect Real-Time System manual (2013) [43].

### 2.7. Statistical Analyses

Statistical analyses on data were carried out using excel. A one-way analysis of variance ANOVA with a 5% level of probability (*p* < 0.05) followed by pairwise mean comparison Tukey test was performed to detect significant differences.

### 2.8. Alkaloid Extraction

Alkaloids were extracted from the 24 *N. papyraceus* samples with methanol and purified following a modified version of the acid-base extraction protocol described previously [44]. For a given sample, 2 g of plant material were ground to a fine powder in liquid nitrogen with a mortar and pestle. The powder was transferred to a 15 mL centrifuge tube. Ten microliters of papaverine (3 µg/mL in water), used as an internal standard for data quantification, and 5 mL of methanol were added to the powder, which was then kept at 50 °C for 2 h. The tube was vortexed briefly and then centrifuged at 7000 g for 2 min. The supernatant (crude extract of alkaloids) was collected in a new 15 mL centrifuge tube and allowed to evaporate until approximately 1.5 mL remained, which was transferred to a 1.7 mL microcentrifuge tube and allowed to evaporate to dryness. The dry crude extract of alkaloids was re-suspended in 300 μL of methanol and subjected to acid-base extraction. First, the sample was acidified with H_2_SO_4_ (2% *v/v* in water). Organic impurities were removed by washing twice with chloroform. Next, alkalization was achieved using ammonia. The purified alkaloid extract obtained was dried under N_2_ gas and finally solubilized in 300 μL of methanol.

### 2.9. High-Performance Liquid Chromatography

The alkaloid extract samples were analyzed by reversed-phase HPLC with photodiode array (PDA) detector using the Prominence-i LC-2030C system (Shimadzu, www.ssi.shimadzu.com). Separations were performed at a flow rate of 0.5 mL/min using a Kinetex C18 column (150 × 4.6 mm, 5 μm particle size; Phenomenex, www.phenomenex.com). Gradient elution was carried out using a 1% aqueous ammonium acetate solution with pH 5 and 100% acetonitrile. The ratio of the ammonium acetate solution to acetonitrile was: 90:10 for 11 min, 69:31 for 4 min, 30:70 for 1 min, 10:90 for 5 min and 90:10 for 2 min. Compounds were monitored at 280 nm.

The average retention time (Rt) and absorption maxima (λ_max_) were determined for the AAs lycorine, galanthamine and narciclasine serving as standards, and for the alkaloid papaverine serving as internal standard for the relative quantification of the concentration of detected compounds (Figure A1; Table A3).

Hundred microliters of the alkaloid extract of each plant sample were injected into the HPLC column. The absorbance value of each compound detected in each sample was normalized to the absorbance value of the internal standard papaverine in that sample (Table A4). Compounds having a normalized absorbance value below 0.20 were discarded. The absorbance of a compound represents its concentration. Furthermore, the relative concentration of lycorine was determined for each plant sample by dividing its absorbance by the total absorbance of all compounds in the same sample.

### 2.10. Accession Numbers

The sequences described in this paper have been deposited in the NCBI Sequence Read Archive under the accession number SRR6041662 (https://www.ncbi.nlm.nih.gov/sra/?term=SRR6041662). Gene transcript sequences were deposited in GenBank with the following accession numbers for nucleotide sequences: tyrosine decarboxylase 1 (MF979854), tyrosine decarboxylase 2 (MF979855), phenylalanine ammonia lyase 1 (MF979856), phenylalanine ammonia lyase 2 (MF979857), phenylalanine ammonia lyase 3 (MF979858), cinnamate 4-hydroxylase 1 (MF979859), cinnamate 4-hydroxylase 2 (MF979860), coumarate 3-hydroxylase (MF979861), 4-coumarate-CoA ligase 1 (MF979862), 4-coumarate-CoA ligase 2 (MF979863), 4-coumarate-CoA ligase 3 (MF979864), 4-coumarate-CoA ligase 4 (MF979865), hydroxycinnamoyltransferase 1 (MF979866), hydroxycinnamoyltransferase 2 (MF979867), hydroxycinnamoyltransferase 3 (MF979868), norbelladine 4’-*O*-methyltransferase (MF979869), noroxomaritidine synthase 1 (MF979770), noroxomaritidine synthase 2 (MF979871), noroxomaritidine/norcraugsodine reductase 1 (MF979872), noroxomaritidine/norcraugsodine reductase 2 (MF979873), noroxomaritidine/norcraugsodine reductase 3 (MF979874) and histone (MF979875).

## 3. Results and Discussion

### 3.1. RNA-Seq of *Narcissus papyraceus* Bulb

To study gene expression in non-model plants such as Amaryllidaceae, for which genomic information is lacking, a transcriptome must be de novo assembled. To achieve this, RNA-Seq of the *N. papyraceus* bulb was first performed. The bulb was chosen because of its high alkaloid content. For example, Viladomat et al. (1986) [34] found that the bulb of *N. assoanus* accumulated a large amount of alkaloids, while Kreh (2002) [32] reported that the concentration of galanthamine was relatively high in the bulb of *N. pseudonarcissus* (L.) cv. Carlton. In a more recent study by Shawky et al. (2015) [33], in which nine different samples (different parts at different developmental stages) of *N. papyraceus* were analyzed, alkaloids were found to be most abundant in the bulb.

RNA was extracted from *N. papyraceus* bulbs and screened for sufficient quality and quantity prior to cDNA library creation and deep sequencing. A total of 70,409,091 raw paired reads were obtained, which were trimmed to give 64,038,268 surviving paired reads that corresponds to 91% of the initial raw reads (Table 1; Table A5). A total of 8,945,044 paired reads were obtained after the normalization step, corresponding to 14% of the initial raw reads. The normalized reads were used to assemble the transcriptome generating 148,563 transcripts (or unigenes) with a length distribution N50 of 1360 bp (Figure A2). For identification of the transcripts, they were aligned against the uniprot_sprot.trinotate_v2.0.pep protein database. BLAST annotation yielded 8866 transcripts with an average length of 2043 bp (Table 1; Table A5). Thus, the remaining 139,697 transcripts represented a large number of transcript sequences showing no similarity to known genes. They appeared to represent transcripts of uncharacterized genes or sequences specific to *N. papyraceus*. Altogether, we concluded that a good quality transcriptome was developed with a high number of surviving paired reads and full-length assembled transcripts.

To identify specific transcripts encoding enzymes involved in the AA biosynthesis in our assembled transcriptome, we performed local BLASTx analyses (Table 2). Several transcript variants of orthologous genes were identified. In a study conducted by Singh and Desgagné-Penix (2017) [25], two *TYDC* transcript variants were identified in *N. pseudonarcissus*. We observed that *N. papyraceus*’s TYDC1 and TYDC2 together share 57% homology of the amino acid sequence with *N. pseudonarcissus*’s TYDC1 and TYDC2; *N. papyraceus* TYDC1 is 90% homologous to *N. pseudonarcissus* TYDC1 and *N. papyraceus* TYDC2 is 94% homologous to *N. pseudonarcissus* TYDC2. The BLASTx analysis further reported that *N. papyraceus* TYDC1 shared 70% homology with opium poppy’s (*Papaver somniferum*) TYDC2, which is involved in the synthesis of benzylisoquinoline alkaloids, and that *N. papyraceus* TYDC2 was 74% homologous to rice’s (*Oryza sativa*) TYDC1 (Table 2).

*PAL* genes have also been reported in various plant species including the Amaryllidaceae *Lycoris radiata* and *N. pseudonarcissus* in which two transcript variants were identified in each species [25,31,45]. BLASTx searches for *PAL* genes in the assembled *N. papyraceus* bulb transcriptome led to the identification of three full-length transcript variants (Table 2). Similarly, several transcript variants were also identified for *C4H*, *4CL* and *HCT* (Table 2). For gene transcripts “specific” to AA biosynthesis, we identified one transcript variant of *NBS*, one of *N4OMT*, two of *CYP96T* and three of *NorRed* (Table 2). Altogether, we were able to identify full-length transcripts, and several variants, with expect values ≤ 0, corresponding to genes encoding biosynthetic enzymes involved in AA production.

Comparative FPKM digital expression of the *N. papyraceus* bulb transcriptome indicated that the “AA-specific” gene transcripts, such as those of *N4OMT*, *CYP96T* and *NorRed*, were more abundantly expressed than transcripts encoding enzymes involved in the “AA-precursor” biochemical reactions including those of *TYDC*, *PAL* and others (Table 2). In addition, *TYDC* and *PAL* had similar levels of expression (Table 2). They may thus be coordinately regulated since both these genes encode enzymes catalyzing the first reactions in the AA biosynthetic pathway (Figure 1). Furthermore, the digital expression profile of the *N. papyraceus* bulb is comparable to that reported for the *N. pseudonarcissus* bulb [25].

### 3.2. Gene Expression Analysis with RT-qPCR

The relative expression levels of the genes analyzed are shown in Figure 3. The results indicate that the transcripts exhibited different expression kinetics that can be described as early (stage 1), mid (stages 2–4) and late (stages 5–6) responses. The fold-expression varied from relatively low (<2-fold) to high (>4000-fold), compared to each other. Looking at the expression levels of any gene in all the 24 *N. papyraceus* samples, it is obvious that its expression differs among the various parts and changes as the plant grows and develops. For example, the physiological conditions of an organism, as well as environmental stimuli, have long been known to affect the transcriptional regulation of PAL [46]. The same may, most likely, be true for all the other AA biosynthetic genes, which can explain the observed spatial and temporal differences in gene expression. Obviously, as time goes on, a plant adapts to its changing needs and responds to changes in its environment, to varying extents, in the various parts of the plant.

The biosynthetic genes *PAL*, *C4H*, *4CL*, *HCT* and *C3H* (Figure 1) are involved in phenylpropanoid biosynthesis. In other words, 3,4-DHBA is a product of the phenylpropanoid biosynthetic pathway [25]. This pathway is responsible for the production of a large number of specialized metabolites including coumarins, hydroxycinnamates and lignins [47,48]. These compounds play essential roles in plant growth and development and plant–environment interactions [46]. Being one of the first biosynthetic genes of the pathway, PAL is responsible for converting the amino acid phenylalanine, a product of primary metabolism, to trans-cinnamic acid (Figure 1). Thus, PAL resides at a metabolically important position, linking primary metabolism to specialized metabolism [49]. The same is true for the biosynthetic gene *TYDC,* which encodes TYDC that is responsible for transforming the amino acid tyrosine (primary metabolite) to tyramine. In addition to AAs, TYDC is involved in the biosynthesis of several other types of specialized metabolites [27] including benzylisoquinoline alkaloids [50] and hydroxycinnamic acid amides of tyramine (HCAATs) [51]. Morphine and codeine are the best-known examples of benzylisoquinoline alkaloids produced by the opium poppy. The production of HCAATs has been associated with plant stress responses. Following oxidative polymerization, HCAATs are believed to reinforce cell walls rendering them less susceptible to penetration by pathogens. Since *PAL* and *TYDC* are genes encoding the first biosynthetic enzymes involved in the production of many types of specialized metabolites, they have attracted considerable attention and are targets for metabolic engineering [52].

Several transcript variants of *PAL* exist since PAL is encoded by a multi-gene family in plants [53]. For example, there are four variants of *PAL* in Arabidopsis, five in poplar and nine in rice [54]. We analyzed the expression of two transcript variants of *PAL* in our study of *N. papyraceus* (Figure 3). At each developmental stage, *PAL1* was expressed at generally similar levels in all parts. *PAL2*, for its part, was expressed the highest in the bulb, at almost all stages, followed by the stem, indicating that PAL2 is more specifically involved in the synthesis of phenylpropanoids in these two parts. This profile of *PAL2* expression has also been reported for *PAL2* of *N. pseudonarcissus* ‘King Alfred’ [25]. The fact that the two transcript variants are expressed at different levels in the various *N. papyraceus* plant samples suggests that they are involved in the phenylpropanoid pathway to different degrees for the production of various specialized metabolites in the plant.

Such differential spatial and/or temporal expression of variants of *PAL*, associated with specific metabolic activities, has also been reported for other plants species. For example, in a study conducted on *PAL* in another Amaryllidaceae species, *Lycoris radiata*, Jiang et al. (2013) [31] identified two transcript variants of *PAL,* which differed in their expression in different parts of the flower at the blooming stage of the plant; the expression of *LrPAL2* was always much higher than that of *LrPAL1*. Singh and Desgagné-Penix (2017) [25] also noted different expression profiles for two *PAL* variants in five parts of *N. pseudonarcissus* ‘King Alfred’ at the flowering stage. In both studies, the authors suggest that the *PAL* variants encode enzymes that may have distinct functions in different branches of the phenylpropanoid pathway. In an analysis of PAL in quaking aspen (*Populus tremuloides* Michx.), Kao et al. (2002) [55] observed that *PtPAL1* is implicated in the biosynthesis of non-lignin metabolites while *PtPAL2* is involved in lignin formation. Lignins are phenylpropanoid-derived compounds involved in cell wall formation.

Regarding the biosynthetic gene *TYDC*, similar to *PAL*, it is encoded by multiple genes. For example, eight copies of the *TYDC* gene have been detected in the opium poppy [57], whereas four similar genes encode *TYDC* in parsley [58]. In our study of *N. papyraceus*, when comparing the expression profiles of *TYDC1* and *TYDC2*, the most interesting feature is that they are practically the same in the first three developmental stages (Figure 3). This may be a sign that these two transcript variants are regulated in a similar manner early in the life of the plant. However, differences in gene expression levels between *TYDC1* and *TYDC2* appear in the later stages of 4–6.

Different expression profiles for different *TYDC* transcript variants have also been observed in the mature opium poppy [59]. They noted that *TYDC1*-like genes were predominantly expressed in the roots and, to a much lesser extent, in the stem. The opposite was true for *TYDC2*-like genes. In the opium poppy, *TYDC* is involved in the synthesis of benzylisoquinoline alkaloids. Facchini and De Luca (1995) [59] analyzed the benzylisoquinoline alkaloids sanguinarine and morphine, each of which is synthesized by different branch pathways further downstream in the benzylisoquinoline alkaloid biosynthetic pathway [50]. The presence of high quantities of sanguinarine in the roots, where *TYDC1*-like genes were highly expressed, and the lack of detectable quantities of sanguinarine in the aerial parts, where *TYDC1*-like genes were lowly expressed, led the authors to believe that *TYDC1*-like genes were particularly involved in the production of sanguinarine and other related alkaloids. On the other hand, the abundance of morphine in the aerial parts together with the high levels of *TYDC2*-like transcripts in the stem, led the authors to suggest that *TYDC2*-like genes may be implicated in the synthesis of morphine and other closely related alkaloids. Based on these observations, the authors propose that *TYDC1*-like genes may be coordinately regulated with genes present in the branch pathway involving sanguinarine biosynthesis, whereas *TYDC2*-like genes may be coordinately regulated with genes present in the branch pathway involving morphine biosynthesis. Such coordinated regulation of expression between different genes involved upstream and downstream in a biosynthetic pathway may be occurring in the AA pathway as well. Further elucidation of the AA pathway will eventually allow the investigation of whether such a phenomenon is present in the AA biosynthetic pathway.

*HCT* is another gene for which we analyzed the expression levels of two transcript variants (Figure 3). Like for *PAL* and *TYDC*, there are considerable differences between the expression level profiles of the *HCT* variants. We assume that, like for *PAL*, the different variants of *HCT* are involved to varying extents in the phenylpropanoid pathway. *HCT* has been reported to be quite active in lignin formation [60,61]. Therefore, maybe the relatively higher expression of *HCT1* during stages 3–6 could be due to a possibly greater production of lignins during these stages and that *HCT1* in *N. papyraceus* is perhaps more involved in lignin formation than *HCT2*.

*C4H* and *C3H* are two biosynthetic genes in the phenylpropanoid pathway, which code for cytochrome P450 enzymes. Curiously, both genes display a similar expression pattern for stages 2–6 (Figure 3). It, thus, seems that these two genes are being coordinately regulated as the plant is growing and developing. Regarding *C4H*, Fock-Bastide et al. (2014) [62] reported that its expression level in *Vanilla planifolia* pods was highest during the earlier stages of its maturation, but declined during the later stages. This expression profile resembles that observed by us for *N. papyraceus* in that *C4H* expression is high during the 1st and 2nd developmental stages (Figure 3). During stages 3–5, the level of expression of *C4H* remains constant even though the *N. papyraceus* plant continues to grow and develop during these stages. Phimchan et al. (2014) [63] also noted that *C4H* activity remained constant in various cultivars of *Capsicum* regardless of fluctuations in capsaicinoid accumulation in these cultivars.

Unlike the genes discussed so far, *N4OMT* is specific to AA biosynthesis. N4OMT is responsible for the methylation of norbelladine, the compound from which all AAs are derived (Figure 1). Kilgore et al. (2014) [30] and Singh and Desgagné-Penix (2017) [25] analyzed the expression of *N4OMT* in the *Narcissus* species at the flowering stage in different parts: Bulb, leaves, flowers and/or roots and stem. Kilgore et al. (2014) [30] noted that the highest expression was in the bulb, followed by the flowers in close second, while it was barely expressed in the leaves. Singh and Desgagné-Penix’s (2017) [25] results demonstrated that the highest expression was, once again, in the bulb but very low in all the other parts. In our results of *N4OMT’s* expression in *N. papyraceus* at developmental stage 5 (flowering stage; Figure 3), we observed that the expression in the bulb was slightly higher than that in the other tissues, and the lowest expression was in the leaves. Our results abound in the same way as those presented above with smaller differences. One reason for the differences in the results of *N4OMT* expression between our and the other two studies may be that the same *Narcissus* species was not analyzed in each study.

### 3.3. Alkaloid Profiling

Representative chromatograms of the alkaloid extract of certain *N. papyraceus* samples are shown in Figure A3. Among the various compounds detected in all the *N. papyraceus* samples, 28 presented interesting features and were selected for analysis (Table A4). Some of these were detected in all or almost all the samples, while others were present in particular parts throughout the life of the plant. The most striking findings are the extremely high and very high concentration of compounds in the leaves of stages 2 and 5, respectively (Figure 4A). This is specifically due to the six compounds with an average Rt ranging from 6.43 to 7.20, which are present at surprisingly high concentrations (Table A4). Looking at the distribution of the compounds among the different parts of the plant, the leaves and flowers had the highest abundance of compounds, whereas the bulb, the lowest (Figure 4A,B). A similar finding was reported in a recent study of AA biosynthesis in *N. pseudonarcissus* ‘King Alfred’ using LC–MS in that leaves had the highest concentration of AAs [25].

By comparing the 28 selected compounds (Table A4) with the results obtained for the AA standards (Figure A1), the following conclusions can be made. The compound with an average Rt of 4.22 min (average λ_max_ of 221 and 285 nm, not shown) corresponds to lycorine since these values are in agreement with those of the lycorine standard. Contrary to lycorine, galanthamine was not detected in any of the samples. Narciclasine also does not seem to be present even though there was a compound detected in almost all of the samples with an average Rt of 7.69 min, which is very close to that obtained for narciclasine (7.70 min). However, the mean λ_max_ values (221 and 283 nm, not shown) of this compound differ considerably from those of narciclasine (251 and 304 nm). Therefore, among the three AA standards, only lycorine was found to be detected in *N. papyraceus*.

Lycorine was detected in each plant sample and had a high concentration in most samples (Figure 4C; Table A4). This is in accordance with the findings reported by Shawky et al. (2015) [33] and confirmed by Tarakemeh et al. (2019) [64] who found that lycorine was the predominant alkaloid in *N. papyraceus* with no/low galanthamine, and with the fact that lycorine is one of the most abundant AAs in the *Narcissus* genus [65]. A reason for this is that lycorine is possibly, phylogenetically, among the oldest AAs in *Narcissus* [66]. Furthermore, lycorine was present at higher concentrations in the leaf samples compared to the bulb samples (Table A4). This is in agreement with gas chromatography-mass spectrometry (GC–MS) results of six ornamental species of *Narcissus* [65]. It is also interesting to note that the fraction of the concentration of lycorine out of the total concentration of heterocyclic compounds in each plant part remains more or less constant as the plant ages (Figure 4C; Table A4). This suggests that a somewhat constant level of production of lycorine occurs in each part of the plant throughout its growth and development.

### 3.4. Comparison of Gene Expression and Compound Content Profiles

To better understand the involvement of the proposed AA biosynthetic genes in the production of the AAs at each developmental stage of *N. papyraceus*, we compared the expression profiles (Figure 3) of the AA biosynthetic genes with the heterocyclic compound content profiles (Figure 4A,B). It should be noted that HPLC is a limited analytical method for identification of AA thus correlations about spatial and temporal pattern of AA accumulation and expression of biosynthetic pathway are observations. Some of the noteworthy observations are described below.

At the 2nd stage, relative to the other stages, *TYDC1*, *TYDC2*, *PAL2*, *C4H*, *C3H* and *N4OMT* all exhibit very high expression in the bulb, considerably lower in the roots, and lowest in the leaves. Therefore, at this stage, during which the plant begins its growth from the 1st stage bulb, the expression of most of the genes remains high in the bulb compared to the roots and leaves. According to this, a great amount of alkaloids would be expected to be produced in the bulb and least in the leaves. However, the opposite is observed in Figure 4A,B where the bulb has the lowest amount of compounds and the leaves have the highest. A plausible explanation for this may be that the biosynthetic enzymes and/or alkaloids produced in the bulb are being transported to other parts of the plant, leading to a decreased amount of alkaloids in the bulb and higher amounts in the roots and leaves. Such transport has been previously described in the opium poppy [67,68,69].

At the 3rd developmental stage, the greater amount of compounds being accumulated in the leaves and flowers is in accordance with the high, or very high, expression of *PAL1*, *4CL*, *HCT1* and *N4OMT* in the leaves or flowers. The surprisingly high content of compounds in the stem, where almost all genes are expressed at very low levels, points, once again, to a possible transport of enzymes and/or alkaloids from, possibly, the bulb and roots to the stem.

At stage 4, the expression of *PAL2* is highest in the flowers, *4CL* is higher in all aerial parts, *HCT1* is highest in the leaves and stem and *TYDC2* is highest in the leaves. This is consistent with the high accumulation of compounds in the aerial parts. There is also an increase of compounds in the roots at this stage, particularly the concentration of lycorine (Figure 4A,C; Table A4). The high concentration of lycorine in the stage 4 roots may be due to high expression levels of many other, yet unknown, genes implicated further downstream in the pathway, more precisely, in the ortho-para’ phenol coupling subgroup, which is specific to lycorine synthesis (Figure 1).

Therefore, as the plant grows and develops, it tends to store the compounds in certain parts rather than others regardless of the level of gene expression. This is possibly due, most likely, to a transport of enzymes and AAs within the plant, as mentioned for the 2nd and 3rd stages. When the plant is young, i.e., at stages 2 and 3, it is clear that there is a greater accumulation of compounds in all aerial parts (leaves, stem and un-blossomed flowers; Figure 4A,B). At stage 4, the leaves and flowers continue to accumulate large amounts of compounds. The presence of a high amount of compounds in the aerial parts is in line with the theory that alkaloids act as defense molecules for a plant [1] as the aerial parts, besides being vulnerable to various pathogens, are also visible and easily accessible to herbivores. Additionally, leaving aside the high concentration of lycorine in the stage 4 roots, the amount of compounds in the roots increased as of stage 4 (Figure 4A,B). This may be because roots can be attacked by pathogens present in the soil [59].

## 4. Conclusions

In this study, we provided the first report of Amaryllidaceae *N. papyraceus* transcriptome using the Illumina HiSeq2000 platform. A large number of transcripts and annotation resources were developed and may prove to be valuable tools for future genetic and genomic studies of *N. papyraceys* and other related species. Moreover, the expression of AA biosynthetic genes, from 24 different samples of different parts at various developmental stages of *N. papyraceus*, was analyzed and compared to alkaloid content for each sample. Correlation was observed between the gene expression levels and the amount of compounds accumulated. The results provide a basis for continued studies to better understand AA’s metabolism. Beyond the biological impact, the data can be used as reference to provide gene sequences for the metabolic engineering and should pave the way for advanced Amaryllidaceae biology and biotechnology to produce valuable alkaloids.

## Figures and Tables

**Figure 1 genes-10-00594-f001:**
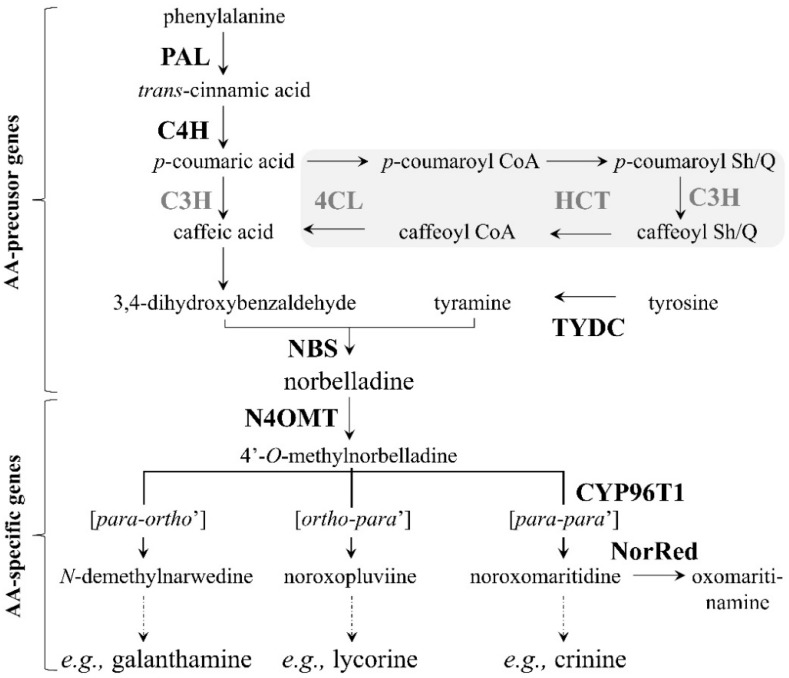
Proposed Amaryllidaceae alkaloid biosynthetic pathway. Abbreviations of enzymes for which corresponding genes have been isolated from Amaryllidaceae plants are written in bold-black font, whereas those isolated from other plants are written in bold-grey font. Reactions in the grey-shaded area represent an alternative route for caffeic acid synthesis. Broken arrows represent more than one biochemical reaction. Abbreviations: PAL, phenylalanine ammonia lyase; C4H, cinnamate 4-hydroxylase (CYP73A1); 4CL, 4-coumarate CoA ligase; HCT, hydroxycinnamoyl transferase; C3H, coumarate 3-hydroxylase (CYP98A3); TYDC, tyrosine decarboxylase; NBS, norbelladine synthase; N4OMT, norbelladine 4′-*O*-methyltransferase; CYP96T1, cytochrome P450 monooxygenase 96T1; NorRed, noroxomaritidine reductase and Sh/Q, shikimate/quinate.

**Figure 2 genes-10-00594-f002:**
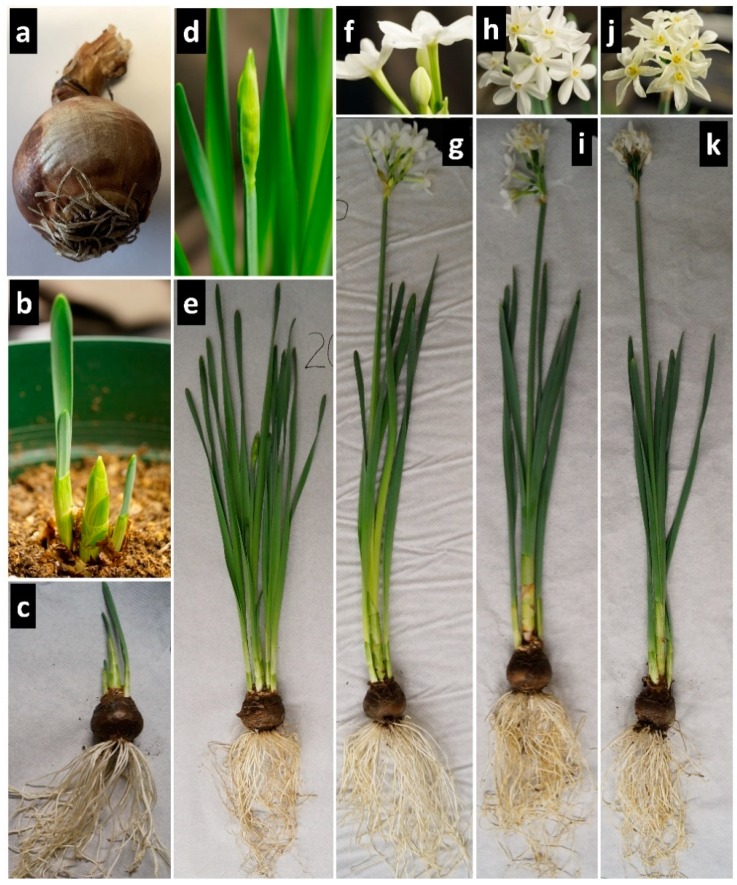
Photos showing the developmental stages of *Narcissus papyraceus* used in this study. (**a**) Stage 1: Unplanted bulb; (**b**–**c**) stage 2: Emergence with presence of young leaves; (**d**–**e**) stage 3: Presence of stems with unopened flower buds; (**f**–**g**) stage 4: Blossoming with emergence of young flowers; (**h**–**i**) stage 5: Flowering with mature flowers and (**j**–**k**) stage 6: Senescence with wilting flowers.

**Figure 3 genes-10-00594-f003:**
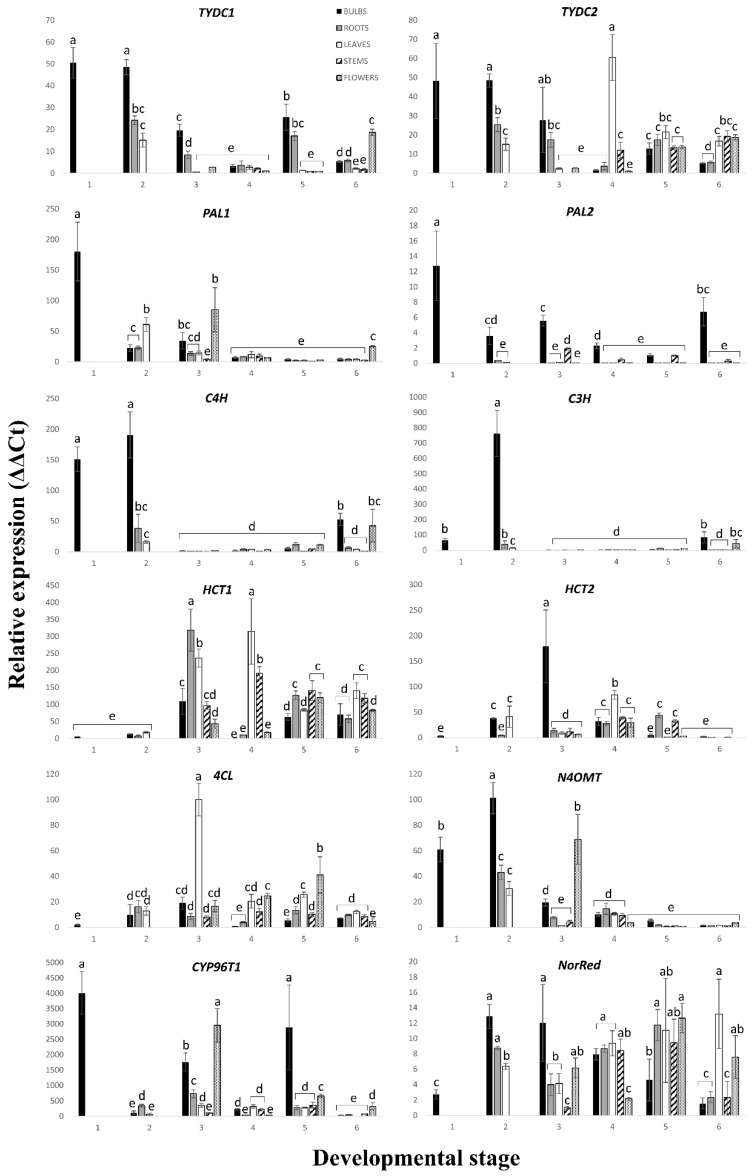
Expression profiles of twelve genes involved in Amaryllidaceae alkaloid biosynthesis obtained by reverse transcription quantitative real-time PCR analysis of different parts (bulb, roots, leaves, stem and flowers) at six developmental stages of *Narcissus papyraceus*. The graphs are plotted using normalized ΔΔC_T_ values scaled to the lowest value. Each graph shows the expression level of the gene indicated above the graph. The gene for histone was used as a reference gene. The gene expression fold change was calculated using the 2^−ΔΔCT^ method [56]. The error bars represent the mean standard deviation of three independent replicates, which were calculated using the method stated in the CFX Connect Real-Time System manual [43]. Abbreviations are *TYDC*, *tyrosine decarboxylase*; *PAL*, *phenylalanine ammonia lyase*; *C4H*, *cinnamate 4-hydroxylase*; *C3H*, *coumarate 3-hydroxylase*; *4CL*, *4-coumarate CoA ligase*; *HCT*, *hydroxycinnamoyl transferase*; *N4OMT*, *norbelladine 4′-*O*-methyltransferase*; *CYP96T1*, *cytochrome P450 monooxygenase 96T1* and *NorRed*, *noroxomaritidine reductase*. The developmental stages are detailed in Figure 2. The bars having the same letter present no significant differences (*p* < 0.05) according to the Tukey statistical test.

**Figure 4 genes-10-00594-f004:**
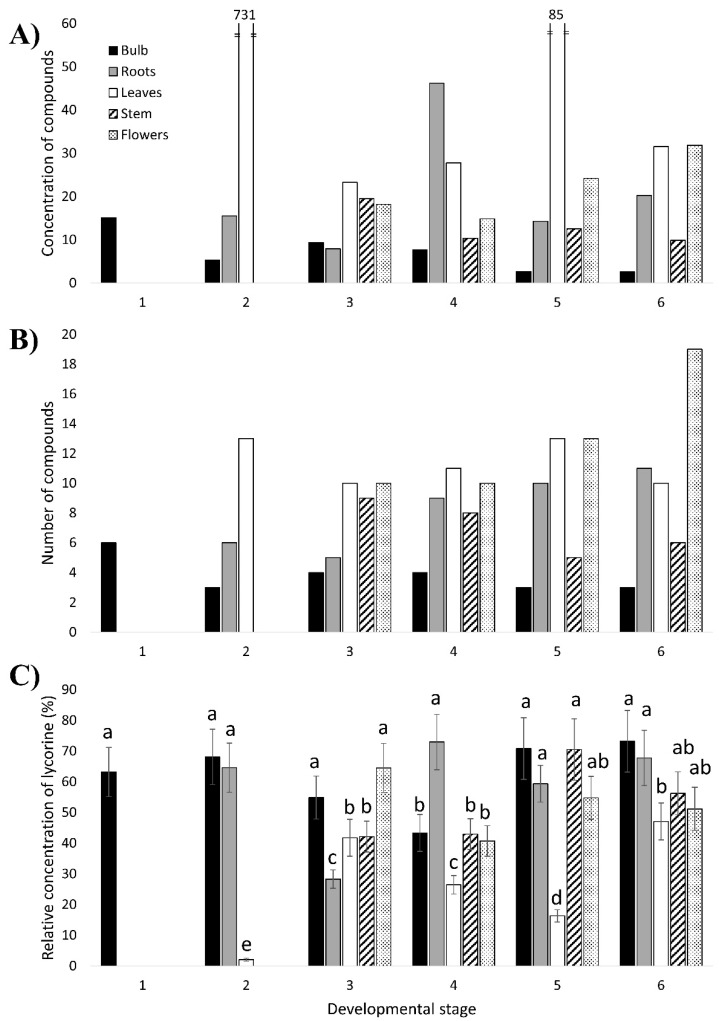
Accumulation profiles of heterocyclic compounds extracted from different parts (bulb, roots, leaves, stem and flowers) at six developmental stages of *Narcissus papyraceus* obtained by HPLC- photodiode array (PDA) analysis. (**A**) The concentration of compounds detected in each plant sample. (**B**) The number of compounds detected in each plant sample. (**C**) Relative concentration of lycorine in each plant sample. This is the ratio of the concentration of lycorine to the total concentration of all compounds in the sample. The extracts having the same letter present no significant differences (*p* < 0.05) according to the Tukey statistical test.

**Table 1 genes-10-00594-t001:** Summary of the transcriptome database generated from one lane of Illumina RNA-Seq of *Narcissus papyraceus* stage 1 bulbs.

	*N. papyraceus*
Total number of raw paired reads ^a^	70,409,091
Total number of surviving paired reads trimmed ^b^	64,038,268
Total number of surviving paired reads normalized ^c^	8,945,044
Number of transcripts ^d^	148,563
Number of components ^e^	86,994
Number of annotated transcripts ^f^	8866
N50 of annotated transcripts (bp)	2043

^a^ Number of paired reads obtained from the sequencer; ^b^ number of remaining paired reads after the trimming step; ^c^ number of remaining paired reads after the normalization step; ^d^ number of transcripts: Trinity has created a list of transcripts representing the transcriptome variants. ^e^ A component is a family of similar transcripts, most likely RNA variants; ^f^ Each transcript has been aligned against the *uniprot_sprot.trinotate_v2.0.pep* protein database using the blastx program from the NCBI BLAST family.

**Table 2 genes-10-00594-t002:** Summary of the biosynthetic genes identified in the *N. papyraceus* transcriptome known to be involved in Amaryllidaceae alkaloid metabolism.

Name	Reads (FPKM)	Length (bp)	ORF (bp)	Top Annotation	Species	Expect Value	Accession Number
*TYDC1*	1881	1641	1536	tyrosine/DOPA decarboxylase 2	*Papaver somniferum*	0	P54769.1
*TYDC2*	1111	2004	1356	tyrosine decarboxylase 1	*Oryza sativa*	0	Q7XHL3.1
*PAL1*	1911	2277	2130	phenylalanine ammonia-lyase 3	*Petroselinum crispum*	0	P45729.1
*PAL2*	1790	2395	2136	phenylalanine ammonia-lyase 1	*Prunus avium*	0	O64963.1
*PAL3*	1790	2395	2136	phenylalanine ammonia-lyase 1	*Prunus avium*	0	O64963.1
*C4H1*	3309	1792	1518	cinnamic acid 4-hydroxylase	*Zinnia violacea*	0	Q43240.1
*C4H2*	3309	1743	1518	cinnamic acid 4-hydroxylase	*Zinnia violacea*	0	Q43240.1
*C3H*	2018	1830	1530	*p*-coumarate 3-hydroxylase	*Arabidopsis thaliana*	0	O22203.1
*4CL1*	1645	1846	1710	4-coumarate:CoA ligase 2	*Oryza sativa*	0	Q42982.2
*4CL2*	1173	1994	1683	4-coumarate:CoA ligase-like 7	*Oryza sativa*	0	Q69RG7.1
*4CL3*	1173	1834	1614	4-coumarate:CoA ligase-like 7	*Arabidopsis thaliana*	0	Q9M0 × 9.1
*4CL4*	1173	2023	1653	4-coumarate:CoA ligase-like 4	*Oryza sativa*	0	Q10S72.1
*HCT1*	2572	1633	1311	hydroxycinnamoyltransferase 1	*Oryza sativa*	0	Q0JBZ8.1
*HCT2*	2572	1807	1416	hydroxycinnamoyltransferase 2	*Nicotiana tabacum*	2 × 10^−96^	Q8GSM7.1
*HCT3*	2572	1612	1308	hydroxycinnamoyltransferase 3	*Oryza sativa*	1 × 10^−105^	Q5SMM8.1
*NBS*	1352	729	480	norbelladine synthase	*Narcissus pseudonarcissus ‘*King Alfred’	1 × 10^−96^	AYV96792.1
*N4OMT*	21722	1062	720	norbelladine 4’-*O*-methyltransferase	*Narcissus aff. pseudonarcissus*	2 × 10^−171^	A0A077EWA5.1
*CYP96T1*	16956	1866	1605	noroxomaritidine synthase (CYP96T1)	*Narcissus aff. pseudonarcissus*	0	A0A140IL90.1
*CYP96T2*	16956	1839	1605	noroxomaritidine synthase (CYP96T1)	*Narcissus aff. pseudonarcissus*	0	A0A140IL90.1
*NorRed1*	6876	1049	810	noroxomaritidine/norcraugsodine reductase	*Narcissus aff. pseudonarcissus*	2 × 10^−169^	AOP04255.1
*NorRed2*	6876	1049	780	noroxomaritidine/norcraugsodine reductase	*Narcissus aff. pseudonarcissus*	1 × 10^−164^	AOP04255.1
*NorRed3*	6876	1049	768	noroxomaritidine/norcraugsodine reductase	*Narcissus aff. pseudonarcissus*	3 × 10^−173^	AOP04255.1
*Histone*	9658	437	410	histone H3.3	*Arabidopsis thaliana*	3 × 10^−95^	P59169.2

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
