# Peer review of "Developmental Regulation of the Expression of Amaryllidaceae Alkaloid Biosynthetic Genes in Narcissus papyraceus"

_genes, 2019, doi:10.3390/genes10080594_

Round 1

Reviewer 1 Report

Hotchandani et al manuscript presents an important piece of work which helps understanding biosynthesis of  Amaryllidaceae alkaloids (AA) - compounds which  have multiple biological effects, and are of interest to the pharmaceutical industry. Authors,  have provided valuable transcriptomic datasets on the regulation of the expression of the genes potentially involved in AA biosynthesis in  various developmental stages of Narcissus  papyraceus. First part of the paper includes RNAseq and qRT-PCR results and those experiments are scientifically quite robust and contain only minor flaws, which are highlighed below and also marked in the attached manuscript:

1. Number of annotated transcripts is quite low. Uncharacterized and species-specific gene would not account for large majority of transcripts! Perhaps an alternative method should be tested for assigning functional annotations to contigs

2. Figure 3: Please indicate statically significant changes. Multiple technical and biological replication should allow to use one-way ANOVA and Tukey's test for example. A simple T-test results would actually be a minimum to show

Second part of the paper however where authors present metabolomics data is not so sound anymore. HPLC-UV method used to detect, characterize and quantify AA is in my opinion not specific enough to support authors main conclusions (see marked manuscript attached)

1.  HPLC with UV detection is not really the most accurate analytical method. There are multiple compounds in the plant extracts which have similar or even identical UV-spectra to the standards used. This is causing confusions as the identity of HPLC-peaks is wrongly assigned. I am afraid the only way of getting robust set of data is to do HPLC-MS where Mass spectrometry would be able to assigned peak IDs with confidence.  I think this is crucial to support any point which authors make about spatial and temporal pattern of AA accumulation and its correlations with the expression of biosynthetic pathway genes. 

2. Figure 4: No statistic done. Please indicate Stadard Deviation or Standard Error at least and indicate significant changes. Figure legend is really small and hard to read. 

Author Response

Point-by-point response to the reviewer 1 ’s comments

We would like to thank reviewer 1 for his thoughtful comments and efforts towards improving our manuscript. In the following, we address each comments. We thank the referee for the careful and insightful review of our manuscript. We address all of the concerns of referee #1 here.

Comments and Suggestions for Authors

Hotchandani et al manuscript presents an important piece of work which helps understanding biosynthesis of  Amaryllidaceae alkaloids (AA) - compounds which  have multiple biological effects, and are of interest to the pharmaceutical industry. Authors,  have provided valuable transcriptomic datasets on the regulation of the expression of the genes potentially involved in AA biosynthesis in  various developmental stages of Narcissus  papyraceus. First part of the paper includes RNAseq and qRT-PCR results and those experiments are scientifically quite robust and contain only minor flaws, which are highlighed below and also marked in the attached manuscript:

Responses to Reviewer 1

Number of annotated transcripts is quite low. Uncharacterized and species-specific gene would not account for large majority of transcripts! Perhaps an alternative method should be tested for assigning functional annotations to contigs

This a good point, indeed an alternative method could be tested to increase the number of annotated transcripts. The sequencing data are available so that several additional bioinformatic studies can be performed. For the purpose of this manuscript, we focused on the twelve identified transcripts for alkaloid biosynthesis and validation of their expression across the 24 different samples.

Figure 3: Please indicate statically significant changes. Multiple technical and biological replication should allow to use one-way ANOVA and Tukey's test for example. A simple T-test results would actually be a minimum to show

 A Tukey test was performed and added in the revised manuscript.

Second part of the paper however where authors present metabolomics data is not so sound anymore. HPLC-UV method used to detect, characterize and quantify AA is in my opinion not specific enough to support authors main conclusions (see marked manuscript attached)

HPLC with UV detection is not really the most accurate analytical method. There are multiple compounds in the plant extracts which have similar or even identical UV-spectra to the standards used. This is causing confusions as the identity of HPLC-peaks is wrongly assigned. I am afraid the only way of getting robust set of data is to do HPLC-MS where Mass spectrometry would be able to assigned peak IDs with confidence.  I think this is crucial to support any point which authors make about spatial and temporal pattern of AA accumulation and its correlations with the expression of biosynthetic pathway genes. 

We agree with Reviewer 1, HPLC-UV is not a good method for identification of compounds. Fine studies were performed by others groups (Shawnky et al. 2014; Tarakemeh et al. 2019) support the presence of alkaloids in N. papyraceus. These studies are reported in the revised manuscript. We did not want to repeat these studies. We revised to manuscript to eliminate conclusion based on identification alkaloids in out studies. We kept the data on lycorine, galanthamine and narciclasine since we had those authentic standards to compare with.

Figure 4: No statistic done. Please indicate Stadard Deviation or Standard Error at least and indicate significant changes. Figure legend is really small and hard to read. 

Statistic on Figure 4 C was done and added in the revised manuscript.

Reviewer 2 Report

Remarks:

> lanes 45-51

I suggest to mention about the biotechnological methods based on in vitro cultures, used for the biosynthesis of Amarylidaceae alkaloids. For this I suggest to include the following references:

1. Ptak A. et al. 2017. Elicitation of galanthamine and lycorine biosynthesis by Leucojum aestivum L. and L. aestivum ‘Gravety Giant’ plants cultured in bioreactor RITA®. Plant Cell Tissue and Organ Culture 128, 335–345.

2. Pavlov A. et al. 2007. Galanthamine production by Leucojum aestivum in vitro systems. Process Biochemistry 42, 734–739.

 >Lane 176

I think that you can add to the Supplementary files the figure showing extracted RNA on the gel. Besides, 100 ng /μl RNA was used for RT? it seems very little.

> lane 243

Discussion should be written with the capital letter (Results and Discussion)

> lane 250

All names of the plant species (first time in the text) should be followed by the names of the authors who distinguished them e.g. Leucojum aestivum L.

> Table 2

The variety of N. pseudonarcissus should be written with a capital letter, without italics and with the quotation marks: N. pseudonarcissus ‘King Alfred

>Figure 3

In the description of the Y axis, instead of “ddCt” it should be “ΔΔCt”

> lane 436

I suggest to include to the Discussion the following reference:

Tarakemeh, A. et al. 2019. Screening of Amaryllidaceae alkaloids in bulbs and tissue cultures of Narcissus papyraceus and four varieties of N. tazetta. J. Pharm. Biomed. Anal. 172, 230-237.

>Figure 4

Figure 3 is not clear. Please, change it to a more readable one, similarly to figures 4A and 4B. On the Y axis, please use a range of 100%, and on the X axis please show the bars for different organs (bulbs, roots, leaves, stem and flowers) one by another (not stacked) at 6 developmental stages.

> References

The journal names should be abbreviated and written with capital letters e.g. instead of ”Journal of ethnopharmacology” there should be “J. Ethnopharm.” It applies to references number: 3, 4, 6, 9 etc.).

This section needs a revision also due to inaccuracy in writing titles of the publications.

> In the entire manuscript, many references are not numbered (lanes: 138, 248, 249, 250, 278 etc.). Please check it in accordance with the guidelines of the publisher.

> Supplementary Table A1

For the parameter ‘Days’ please, specify the range from to, instead of the average day with fractions (e.g. ±6.4). Please, specify also the replication number for the number and the length of leaves.

Author Response

Point-by-point response to the reviewer 2 ’s comments

We thank reviewer 2 for the careful and insightful review of our manuscript. We address all of the concerns here.

Responses to Reviewer 2

Comments and Suggestions for Authors

Remarks:

> lanes 45-51

I suggest to mention about the biotechnological methods based on in vitro cultures, used for the biosynthesis of Amarylidaceae alkaloids. For this I suggest to include the following references:

Ptak A. et al. 2017. Elicitation of galanthamine and lycorine biosynthesis by Leucojum aestivum L. and L. aestivum ‘Gravety Giant’ plants cultured in bioreactor RITA®. Plant Cell Tissue and Organ Culture 128, 335–345. Pavlov A. et al. 2007. Galanthamine production by Leucojum aestivum in vitro systems. Process Biochemistry 42, 734–739.

This is an excellent suggestion. We added the information on the biotechnological methods based on in vitro cultures used for the biosynthesis of Amaryllidaceae alkaloids to the introduction with the suggested references and additional ones.

 >Lane 176

I think that you can add to the Supplementary files the figure showing extracted RNA on the gel. Besides, 100 ng /μl RNA was used for RT? it seems very little.

The sentence was corrected since 10 μL of 100 ng/μL i.e. 1μg was used for RT.

> lane 243

Discussion should be written with the capital letter (Results and Discussion)

 This was corrected in the revised manuscript.

> lane 250

All names of the plant species (first time in the text) should be followed by the names of the authors who distinguished them e.g. Leucojum aestivum L.

 This was corrected in the revised manuscript such as Narcissus papyraceus Ker Gawl.

> Table 2

The variety of N. pseudonarcissus should be written with a capital letter, without italics and with the quotation marks: N. pseudonarcissus ‘King Alfred

 This was corrected in the revised manuscript.

>Figure 3

In the description of the Y axis, instead of “ddCt” it should be “ΔΔCt”

This was corrected in the revised manuscript.

> lane 436

I suggest to include to the Discussion the following reference:

Tarakemeh, A. et al. 2019. Screening of Amaryllidaceae alkaloids in bulbs and tissue cultures of Narcissus papyraceus and four varieties of N. tazetta. J. Pharm. Biomed. Anal. 172, 230-237.

This is an excellent suggestion. We added the reference to our discussion in the revised manuscript.

>Figure 4

Figure C is not clear. Please, change it to a more readable one, similarly to figures 4A and 4B. On the Y axis, please use a range of 100%, and on the X axis please show the bars for different organs (bulbs, roots, leaves, stem and flowers) one by another (not stacked) at 6 developmental stages.

The figure 4C was modified according to the comment of the reviewer in the revised manuscript.

> References

The journal names should be abbreviated and written with capital letters e.g. instead of ”Journal of ethnopharmacology” there should be “J. Ethnopharm.” It applies to references number: 3, 4, 6, 9 etc.).

The journal names were corrected according to the journal guideline.

This section needs a revision also due to inaccuracy in writing titles of the publications.

> In the entire manuscript, many references are not numbered (lanes: 138, 248, 249, 250, 278 etc.). Please check it in accordance with the guidelines of the publisher.

 The references were numbered in the revised manuscript.

> Supplementary Table A1

For the parameter ‘Days’ please, specify the range from to, instead of the average day with fractions (e.g. ±6.4). Please, specify also the replication number for the number and the length of leaves.

This was corrected in the revised manuscript.
